# Experimental Verification of the Impact of Radial Internal Clearance on a Bearing’s Dynamics

**DOI:** 10.3390/s22176366

**Published:** 2022-08-24

**Authors:** Bartłomiej Ambrożkiewicz, Arkadiusz Syta, Anthimos Georgiadis, Alexander Gassner, Nicolas Meier

**Affiliations:** 1Department of Automation, Faculty of Mechanical Engineering, Lublin University of Technology, Nadbystrzycka 36, 20-618 Lublin, Poland; 2Institute of Product and Process Innovation (PPI), Leuphana University of Lüneburg, Universitatsallee 1, 21335 Lüneburg, Germany; 3Department of Computerization and Robotization of Production, Faculty of Mechanical Engineering, Lublin University of Technology, Nadbystrzycka 36, 20-618 Lublin, Poland

**Keywords:** ball bearings, nonlinear mathematical model, experimental verification, recurrence analysis

## Abstract

This paper focuses on the influence of radial internal clearance on the dynamics of a rolling-element bearing. In the beginning, the 2—Degree of Freedom (DOF) model was studied, in which the clearance was treated as a bifurcation parameter. The derived nonlinear mathematical model is based on Hertzian contact theory and takes into consideration shape errors of rolling surfaces and eccentricity reflecting real operating conditions. The analysis showed characteristic dynamical behavior by specific clearance range, which reflects others in a low or high amplitude and can refer to the optimal clearance. The experimental validation was conducted with the use of a double row self-aligning ball bearing (SABB) NTN 2309SK in which the acceleration response was measured by various rotational velocities. The time series obtained from the mathematical model and the experiment were analyzed with the recurrence quantification analysis.

## 1. Introduction

### 1.1. Background

Rolling-element bearings are basic mechanical parts responsible for transferring the rotational movement in a wide range of mechanical engineering systems [1,2,3,4,5]. The global trend on sustainability and constant product development have improved the bearing’s performance through its high-speed operation [6], reduced friction [7], and reduced noise [8]. The above-mentioned operations are related to the reduction in power losses, which have been imposed by the Environmental Protection Agency (EPA), and the reduction in CO_2_ (carbon dioxide) emissions to the atmosphere. Another parameter which has a strong impact on the vibrational characteristics of a bearing and its lifetime is the radial internal clearance corresponding to the total distance that one ring, a relative of the other ring, can be moved. In this paper, we examine its influence on the bearing’s dynamics and verify it in an experimental way. Such an approach strengthens the conclusions coming from the study of the dynamical systems [9,10]. The term for radial internal clearance is connected to the nonlinear deformations occurring on rolling surfaces, which is why it is worth cross-checking its impact on a bearing’s dynamics and validating it in the experiment. In the next subsection, we briefly present the results of other researchers who have dealt with the impact of radial internal clearance on a bearing’s dynamics. 

### 1.2. State of the Art

Over the years, the impact of radial internal clearance on bearing dynamics has been studied. It has been shown that as a part of a nonlinear equation for elastic deformations it cannot be assigned to the specific characteristic frequency. The effect of radial internal clearance on the dynamics of a balanced horizontal rotor was firstly studied by Tiwari et al. [11], who observed various dynamical solutions, i.e., periodic, sub-harmonic, and chaotic due to specific clearance values. Lioulios et al. [12] combined the clearance effect with the fluctuations of rotational velocity and also found stable and unstable regions due to the internal clearance. Similar research regarding obtaining different dynamical solutions was conducted by Harsha et al. [13] who analyzed the level of nonlinearity caused by specific clearance values. Changqing et al. [14] extended the equation for elastic deformations of the mathematical model by the influence of shape errors in form waviness which have an impact on the contact on rolling surfaces. The influence of clearance by high speeds was studied by Upadhyay et al. [15], who showed the effect of increased clearance on the generation of superharmonics with a higher amplitude and the strength of backward whirl components. In another paper, the radial internal clearance was acknowledged among other factors such as applied load, shape errors, and a number of balls as the factor which has a strong impact on the bearing’s dynamics [16]. The approach to control the radial internal clearance in time was studied by Minmin et al. [17] and compared to the dynamical response for the same type of bearings with a different clearance. The same scientific team investigated the vibration response of the bearing’s radial clearance under the effect of gear meshing and a run-to-failure test to validate the effectiveness of the modulation signal bispectrum-sideband estimator (MSB-SE) for online-clearance monitoring [18]. Noticing the continuous research on the field of radial internal clearance impact on bearing dynamics, we continued our research on finding the optimal bearing’s performance due to radial internal clearance [19,20]. In this case, we compared the results obtained with the derived mathematical model with the experimental data. 

### 1.3. Contribution to the Study

A term referring to the radial internal clearance is the part of the nonlinear equation that defines dynamically changing deformations on rolling surfaces and that states that linear methods related to the characteristic frequencies are not sufficient for its analysis. The introduction of various nonlinearities to the dynamical response requires that the data are treated with one of the nonlinear methods, and that is why the recurrence analysis was applied [21,22]. The recurrence analysis can be divided into the qualitative, graphical method called Recurrence Plots (RPs) and the quantitative method called Recurrence Quantification Analysis (RQA), which both refer to the investigation of nonlinear dynamical systems. Studying strongly stochastic systems provides satisfactory results in the analysis of mechanical systems [23,24,25]. This is why the recurrence analysis was applied for the analysis of a nonlinear acceleration time series of a mathematical model and experimental data.

### 1.4. Remainder

The paper is organized as follows. In Section 2, the derived mathematical model of a double-row self-aligning ball bearing is discussed. The bifurcation analysis with the radial internal clearance as a bifurcation parameter is presented in Section 3. In the following section, the experimental setup is described and in Section 5, the comparison between the results of the mathematical model and the experimental data is conducted. The conclusions are drawn in Section 6. 

## 2. Mathematical Model

Self-aligning ball bearings are not as popular as standard deep groove ball bearings; however, they are widely used in the transport, agricultural, and energy sectors. Their uniqueness is based on misalignment accommodation, low friction and heat generation, and low noise and vibration level. The first of the mentioned features is particularly important from the perspective of the derivation of the mathematical model [16,26,27]; most of the self-aligning ball bearings are double-row and their pressure angle is different from 0°. The derived mathematical model is related to the bearing studied in the experiment, i.e., NTN 2309SK, so its real dimensions were used. The model is based on the Hertzian Contact Theory and included the following assumptions: the influence of shape errors, the shaft’s eccentricity, the variable damping factor, the pressure angle, and the effect of the double row. The mathematical representation will be presented in the following subsections. 

### 2.1. Description of the 2-DOF Mathematical Model of the Self-Aligning Ball Bearing (SABB)

The applied mathematical model is presented in the form of the spring-damper oscillator, in which the assembly inner ring-shaft is treated as one rotating mass and the outer ring is fixed. The derived 2-DOF model of the rolling-element bearing represents the operation of double-row self-aligning ball bearing (SABB) in the x-y plane. The specificity of the self-aligning ball bearing is referred to as its variable pressure angle (Figure 1). Moreover, in the studied bearing the cage is separated on each row, so the assumption of a slight change of radial internal clearance on each row is introduced. Another important assumption is the equal distribution of rolling elements in each row around its circumference with the constant angle, so the angular position of each rolling element is determined in the following way:(1)$$\psi_{i}=\psi_{0}+\overset{t}{\underset{0}{\int }}\omega_{c}\left(t\right)dt$$
(2)$$\psi_{0}=\frac{2\pi \left(i-1\right)}{n}$$
where *ψ*_0_ is the angular position of the first ball, *i* is the angular position of the ball (*i* = 0, 1, …, *n* − 1), *n* is the number of balls, and *ω_c_* is the rotational velocity of the cage. The velocity of the retainer is referred to the design of the bearing (dimensions of the diameter of the rolling elements and the pitch diameter) and the velocity of the shaft: (3)$$\omega_{c}=\frac{\omega_{s}}{2}\left(1-\frac{D}{d_{p}}\right)$$
where *ω_s_* is the velocity of the shaft, *D* is the rolling-element diameter, and *d_p_* is the pitch diameter. The input data for the mathematical model are described in Section 4 referring to the experimental setup.

### 2.2. Shape Errors

The effect of imperfections such as roundness and waviness on the characteristics of the vibrations spectra of ball bearings has been proven and has a significant impact [28] on its dynamical response. In the mathematical model, undulations on the rolling surfaces are described with harmonic functions defined with their number and the highest amplitude on each ring. From the point of the radial internal clearance and the tribological point of view, the roundness should be taken into consideration, given that it can radically impact the contact forces and elastic deformations on rolling surfaces (Figure 2). During the constant operation of the bearing, the quality of rolling surfaces changes due to various factors such as lubrication, subjected forces, and the contamination of metallic inclusions. In the derived mathematical model, we focused on the impact of radial internal clearance on the bearing’s dynamics in a relatively short time of operation, and the above-mentioned factors were not taken into account, except the imperfections created during the manufacturing processes of grinding and super-finish.
(4)$$U_{in_{i}}=U_{in}\sin\left(N_{in}\psi_{0}\right)$$
(5)$$U_{out_{i}}=U_{out}\sin\left(N_{out}\psi_{0}\right),$$
where $U_{in},U_{out}$ are the amplitudes of the highest amplitude of raceway’s surface waviness on the inner and outer ring, respectively, $N_{in},N_{out}$ are the number of undulations on the inner and outer ring, respectively.

### 2.3. Eccentricity

After the bearing’s seating on the shaft and after it is connected to the propelling system, there will be always some shaft misalignment between one end of the shaft and the bore. During the rotation, even small imperfections in misalignment can lead to eccentricity and the rotating mass shifting from the center of rotation with some small distance. The effect could be negligible, but in fact, it has an impact on the dynamics of the whole assembly of the rotor-based system. Its level by different velocities is determined for the mathematical model with an eddy current sensor during the initial tests.

### 2.4. Damping Coefficient

The research has focused on the impact of radial internal clearance on the bearing’s dynamics, and the correlation between RIC and damping coefficient should be taken into account. It was proven, experimentally [29], that there is a linear correlation between the damping factor and the value of the radial internal clearance. In Figure 3, damping factor functions in radial internal clearance and rotational velocity domain are presented. It was also proven that the value of the damping factor decreases with the increase in the radial internal clearance. Additionally, the effect of damping decreases with the increase in the rotational velocity, so this effect is also considered by the determination of the damping factor. The value of damping is based on other papers related to the modeling of rolling-element bearings [30,31] and technical brochures of bearing manufacturers [32]. Its value is specified in Table 1 in the next Section referring to the bifurcation analysis.

### 2.5. Hertzian Contact Theory

The elastic deformations occurring on rolling surfaces are mathematically described with help of Hertz contact law referring to non-adhesive elastic contact. The contact between a specific rolling element and raceway emerges in elliptical contact in the loaded zone and the point contact when there is no load. The dynamical output of a rolling-element bearing is strongly nonlinear due to the number of numerous balls and is strongly nonlinear where there is stronger contact. 

Equation (6) refers to the calculation of dynamically varying elastic contact deformations *δ_i_* for each *i*-th ball specified by its angular position *ψ_i_*. Additionally, the equation considers the effect of radial internal clearance *r_c_*, and shape errors described in Equations (4) and (5):(6)$$\delta_{i}=\delta_{xs}\cos\left(\psi_{i}\right)+\delta_{ys}\left(\psi_{i}\right)-r_{c}-U_{in}\sin\left(N_{in}\psi_{0}\right)-U_{out}\sin\left(N_{out}\psi_{0}\right)\;$$
where *δ_xs_*, *δ_ys_*, are elastic deformations in the *x*, and *y* directions, respectively, and *r_c_* is the radial internal clearance. 

Due to the dynamic change of elastic deformations, the stiffness is also calculated dynamically in each direction. Additionally, the situation of contact and no-contact are defined by a 0–1 function called the Heaviside function *H*(•), and the point contact elastic deformations are powered to 3/2.
(7)$$k_{x}=k_{b}\overset{n}{\underset{i=1}{\sum }}H\left(\delta_{i}\right)\delta_{i}^{\frac{3}{2}}\cos\left(\psi_{i}\right)\;$$
(8)$$k_{y}=k_{b}\sum_{i=1}^{n}H\left(\delta_{i}\right)\delta_{i}^{\frac{3}{2}}\sin\left(\psi_{i}\right),$$
where *k_b_* is the ball stiffness. 

### 2.6. Equations of Motion

The governing equations of motion (Equations (9) and (10)) are obtained by the transformation of the Lagrange equation of 2nd and they allow the dynamical response of the self-aligning ball bearing to be obtained. Additionally, in the equations of motion, the eccentricity of the shaft-bearing system and additional load are considered. In the next section, after the definition of all dimensioned terms, the derived equations of motion are presented that were used to conduct the bifurcation analysis for variable radial internal clearance.
(9)$$m{\ddot{\delta }}_{xs}+c_{x}{\dot{\delta }}_{xs}+\delta_{xs}=F_{x}+e\omega_{s}^{2}\cos\left(\omega_{s}t\right)$$
(10)$$m{\ddot{\delta }}_{ys}+c_{y}{\dot{\delta }}_{ys}+\delta_{ys}=F_{y}+e\omega_{s}^{2}\cos\left(\omega_{s}t\right)$$

## 3. Bifurcation Analysis

The next step of the research was the analysis of the derived mathematical model of the studied bearing and primarily the influence of radial internal clearance on its dynamics. For this case, in Table 1 the dimensional terms referring to the bearing’s design and planned experiment are specified. The value of the radial internal clearance is referred to as its real value in the experiment from 0.5 μm to 50 μm, and in this range, it was possible to control the radial clearance during the experiment. The model was derived in Matlab software and was solved using the Runge–Kutta method corresponding to the ODE45 solver with a tolerance of 0.01 and the same value of the time step. 

In order to check the quantitative and qualitative influence of the clearance on bearing dynamics, a bifurcation analysis was conducted. The idea underlying bifurcation analysis is to follow or continue steady states and periodic solutions in a chosen continuation parameter [33,34]. For this case, the bifurcation analysis with radial internal clearance as the bifurcation parameter was conducted for different rotational velocities of the bearing, i.e., *ω_s_* = {10; 20; 30; 40; 50} Hz, so that the dynamical analysis would be conducted with the mentioned velocities (Figure 4). The dependence of eccentricity and damping factor on the rotational velocity is considered in Table 1, and they were assigned to the specific velocity value. The analysis was conducted for the acceleration time series of deformations in the vertical direction *δ_ys_*.

The results obtained show some regions where the number of solutions for the specific clearance had the smallest value proving the optimal value of clearance. The general shape of the bifurcation diagrams for each rotational velocity had an increasing trend with the increase in the clearance. This phenomenon is related to the bigger freedom of rolling elements for moving toward the vertical or horizontal direction. During the bifurcation analysis, the number of solutions for a specific clearance was studied, without focusing on the type of bifurcation. Nevertheless, referring to the nonlinear character of the mathematical model, four specific dynamical behaviors can be distinguished for marked the values of clearance in Figure 4, i.e., *RIC* = {5.5; 16; 30; 48} μm. For *RIC* = 5.5 μm, the first qualitative change was observed, where the first minima were observed showing the transition between the area of increased friction for small values of clearance and the area of increased clearance, where the rolling elements had more freedom of movement. Another area which could be distinguished was between the *RIC* = 5.5 μm up to 30 μm, where the number of acceleration solutions had a rather equal distribution and corresponded to the normal operation of the bearing. For radial internal clearance equal to 30 μm, there was another qualitative change, where a small number of solutions were obtained. The narrow range of clearances around this value was related to the smoothest operation of the bearing corresponding to the value of the optimal clearance referred to in the first experimental analysis [19,20]. After the area corresponded to the optimal clearance, there was a constant increase in solutions with the increase in the clearance, which refers to the freedom of movement for rolling elements. The shape of the bifurcation diagrams and finding the area of interest was similar for all considered rotational velocities, so the dynamical response was not very dependent on the rotational velocity; however, the response was dependent on the value of the radial internal clearance. This dependence will be discussed in Section 5 for the results obtained from the mathematical model and the experiment. 

Moreover, for the four considered clearances, the standard analysis was conducted in the traditional form presenting time series for the deformations in the vertical direction *δ_ys_* (Figure 5) and drawing orbit and phase plots (Figure 6) for the dynamical output of the bearing. Referring to the time series presented in Figure 5, it can be stated that the dynamical response had a strongly nonlinear character, and the level of deformations oscillated around the set value of the radial internal clearance. 

Additionally, the obtained orbit and phase plots (Figure 6) reflected the same strongly nonlinear character due to the escalation of the contact with the number of rolling elements. Both the orbit and phase plots created a closed orbit, but they were very chaotic, and they did not follow the main orbit. Based on the phase plots, the time series were characterized by multiple periods in it, with Poincare points in different places of the derived trajectory. Referring to the results obtained, in Section 5, the comparative analysis between the mathematical model and the experiment is conducted. 

## 4. Test Rigs and Experiment

In the experimental part, two test rigs were used, i.e., an automated system for measuring the radial clearance after mounting (Figure 7) [35,36], and the test rig used for the dynamical tests of bearings (Figure 8) [36]. For the tests, 10 self-aligning ball bearings with a conical bore NTN 2309SK were prepared (Figure 9). It is worth underlining that the tested bearings were taken from the serial production, and they were not prepared for the purpose of the planned experiment. 

The idea of conducting the experiment was to find the correlation between the mathematical model and to check the influence of the radial internal clearance on the dynamics of the ball bearing. It is essential that the main assumptions of the experiment are provided and its details have been discussed in depth in previous papers regarding the experimental part [19,20]. Referring to the experiment and time series processing, the following experiment stages were defined for each studied bearing:Radial internal clearance reduction with axial nut;Radial internal clearance measurement;Bearing’s mounting in the plummer block with pre-defined clearance;Setting operational for one of rotational velocities from 10 to 50 Hz;Acceleration measurements for 10 min;Signal normalization for time series comparison between cases;Signal filtering with frequency range of 0–100 Hz.

The above-described procedure was retaken for all the studied bearings. During the experiment, for each rotational velocity, the level of the shaft’s endplay was measured with the eddy current distance sensor eddyNCDT 3300. The determined level of the system’s influence on the bearing’s dynamics was previously specified in Table 1. For the data acquisition, the professional software was used provided by IFM company (VES004—V2.16.02), in which it was possible to collect the data of the bearing’s acceleration. During the dynamical tests, the data were collected with the highest possible sampling frequency equal to *f_s_* = 1562.5 Hz to ensure that the information about the dominating frequencies in the spectra was not lost and that the level of the radial internal clearance could not be assigned to the characteristic frequency due to nonlinear contact. After obtaining the mathematical model and the experimental acceleration time series, in the next Section, the comparative analysis was performed with help of a non-linear measure of complexity.

## 5. Recurrence Analysis for the Mathematical Model and Experimental Data

After solving the mathematical model and finishing the designed experiment, the comparative analysis was conducted with help of recurrence analysis [21,37]. The advantage of applied recurrence analysis is its sensitivity to noise and the nonlinear character of the studied time series, which were obtained through mathematical model and in the experiment. Moreover, another advantage of the method is that it is usually used for the analysis of short-time series. This property was particularly important for the experimental data, as the radial internal clearance can change slightly in some ranges during the experiment. The comparative analysis of obtained data was conducted in Matlab software with Cross Recurrence Plot (CRP) Toolbox provided by Potsdam Institute for Climate Impact Research [38]. In the following subsections, a brief description of the main assumptions for the recurrence analysis and its application are provided. 

### 5.1. Recurrence Analysis

Recurrence is one of the properties characterizing dynamical systems and can be defined as the system’s return to the same dynamical state after some time. When the phase space trajectory of the dynamical system is close enough to the same area in the phase space, then two points in the close distance are in recurrence. The distance matrix (Equation (11)) is defined by dynamic states with ones (recurrence) and zeros (no recurrence), that can be expressed mathematically in the following way:(11)$$R_{i,j}^{\varepsilon }=H\left(\varepsilon -\left\|\{x_{i}\right\}-\left\{x_{j}\}\right\|\right),\;i,j,\ldots ,\;N$$
where *ε* is the threshold distance, *i*, *j* are corresponding time of time series, and $\left\|\{x_{i}\right\}-\left\{x_{j}\}\right\|$ is the norm defining the recurrence matrix. 

The recurrence analysis is divided into two main methods, i.e., the graphical qualitative method called Recurrence Plots (RP) [37,39] expressing the periodic/non-periodic character of analyzed dynamics and a quantitative method called Recurrence Quantification Analysis (RQA) [40,41] referring to diagonal and vertical lines, recurrence time, and the probability of structures observed in recurrence matrix. For the studied data, we focused only on the level of three chosen recurrence quantificators for the mathematical model and experimental data. 

### 5.2. Phase-Space Reconstruction

According to Takens theorem [42], three missing coordinates must be found to reconstruct the state of the system in delayed coordinates, i.e., time delay—*τ*, embedding dimension—*m*, and threshold—*ε*. After its definition, the dynamical state of the system after reconstruction is presented in the form of a time-delayed vector in the following form (Equation (12)) [43]:(12)$$s_{n}=\left(s_{n-\left(m-1\right)\tau },s_{n-\left(m-2\right)\tau },\ldots ,s_{n-\tau },s_{n}\right)\;$$
where *s_n_*—scalar observation of fixed sampling time. 

To find the first parameter of time delay *τ*, the mutual information (MI) function is applied [44]. The approach is the quantification between the studied time series and the delayed one (shifted in time), and the final value of the time delay applied during the phase-space reconstruction is the first minimum for the mutual information function. 

The next phase-space reconstruction parameter, i.e., embedding dimension *m* can be found with help of the False Nearest Neighbors (FNN) function [45]. Its main assumption is related to the detection of neighboring points at a close distance to each other in the embedding space. The value of the embedding dimension is determined by the first meeting zero in the FNN function. 

The last parameter is the threshold *ε* corresponding to the radius of the trajectory in the phase-space. Its proper selection is a topic of discussion, and the criterion taken for its selection is based on the level of recurrence rate (RR) equal to 5% [46]. Then, the threshold is adjusted to the density of recurrence points in recurrence plots. 

Following the above-described assumptions, the recurrence analysis was applied to the acceleration time series obtained from the mathematical model and experimental data. During the analysis of the mathematical model data, for the analysis of each case, one time series was studied consisting of 3000 data points (the same number of data points was considered for the experimental time series). For the analysis, the part of the time series when the system stabilized and was periodically repeating was taken,. In the case of the experimental data, the situation was much more demanding due to the number of frequencies in the spectra and the buried noise in the signal. That is why, for the experimental approach, before the calculation of RQA, the minima of the embedding dimensions were found as well as the mean value of the time delay. While the embedding dimension was set at a constant for all cases, the time delay was changed during the considered experimental cases. The phase-space reconstruction parameters found are presented in Table 2.

### 5.3. Comparison of the Results and Discussions

Referring to the bifurcation analysis and the impact of radial internal clearance on bearing dynamics, the comparative analysis between the derived mathematical model and the experiment is presented. For the analysis, three recurrence quantificators were chosen, namely, Entropy, Trapping Time, and Average line length of the diagonal line, that characterized the high sensitivity to change of the radial internal clearance and rotational velocity [20]. In Figure 10, the results of the mentioned quantificators for the response of the mathematical model to different radial internal clearances and different rotational velocities are presented. The mathematical representation of the mentioned quantificators is presented in Equations (13)–(15) with its short definition: Entropy *ENTR*—is the measure of the distribution of the diagonal segments. It reflects the complexity of the recurrence plot regarding the diagonal lines:
(13)$$ENTR=-\overset{N}{\underset{l=l_{min}}{\sum }}p\left(l\right)ln\left(p\left(l\right)\right)\;$$

Trapping Time *TT*—denotes the average length of the vertical structures at the recurrence plot:


(14)
$$TT=\frac{\sum_{v=v_{min}}^{N}vP\left(v\right)}{\sum_{v=v_{min}}^{N}P\left(v\right)}$$


Average line length of the diagonal line *L*—shows that a part of the phase-space trajectory is at a close distance during *l* time steps to another part of the phase-space trajectory in a different time:


(15)
$$L=\frac{\sum_{l=\mu }^{N}lP\left(l\right)}{\sum_{l=\mu }^{N}P\left(l\right)}$$


Beginning with the results of entropy, most of the rotational velocities (Figure 10a) reflected the shape of the bifurcation plots. Given that entropy refers to the uncertainty of the bearing’s response, its highest value was obtained for small clearances with clear fluctuations and a smooth increase by big values of clearance. Non-smooth fluctuations of entropy for small clearances are referred to as the small distance between rolling surfaces and intermittent contact and non-contact phenomena. On the other hand, the entropy increased with the increase in the clearance after the second minima, which is referred to as a higher freedom of the rolling elements. Moreover, owing to entropy, two minima were observed at bifurcation plots around *RIC* = 5.5 μm and *RIC* = 30 μm. It is worth noticing that there was an increase in sensitivity to the entropy with the increase in the rotational velocity of bearing, then the transition between consecutive clearances had a smooth character. 

For trapping time (Figure 10b), fluctuations and increased values were observed for a range of small clearances, which was reflected in an increased level of friction and contact. The trapping time refers to the length of vertical lines and level of nonlinearity in the system, which increases due to the contact of rolling elements with raceways. Moreover, the trapping time is related to the the system’s “trapping” in a specific dynamic state and switching to another. That is why for the small clearances mentioned, the quantificator is sensitive and switches between the contact and non-contact solution.

The average length of the diagonal line gives information on the stability and predictability of the dynamical system. In this case, a strong relationship between the rotational velocity and its level was observed. The dynamical response of the bearing has a higher level for lower velocities when the velocity has a stronger impact on the bearing’s dynamics than clearance. Nevertheless, the response of *L* changes in the same way in the clearance function as was observed in the bifurcation plots.

The next step of research was the comparison of mathematical model results with the experimental results discussed in the previous paper [18]. Regarding the experiment, the results of the three mentioned recurrence quantificators are presented (Figure 11) for the first tested bearings. During the experiment it was impossible to conduct the research the same way as in the bifurcation analysis; therefore, precisely six values of the radial internal clearance were set in the bearing and the dynamical test was conducted by five rotational velocities as in the mathematical model. Referring to the experiment, the minima of recurrence quantificators were observed in the range of *RIC* = (25–32) μm, which was observed in the bifurcation plot. In Figure 11, there are two additional cubic functions fitted to the mean value of each quantificator by specific clearance. The blue line refers to the whole range of clearances; however, due to the observed minima, the red line corresponded to the middle range of clearances.

The mentioned observation was reflected in the values of the recurrence quantification analysis, which is part of the discussion.

Beginning with the results of entropy (Figure 11a), its level was similar to the results of the mathematical model and the same as the distribution of recurrence quantificators (Figure 11a). It must be added that the error bars visible in Figure 11 consisted of around 300 samples corresponding to a number of windows in the acceleration time series. Similar to the mathematical model, the fluctuations were observed for small values of the clearance, which was observed for *RIC* = 8 μm. The characteristic minima for the optimal clearance was observed for both the mathematical model and for the experimental data.

Referring to the results of the trapping time (Figure 11b), its results were similar to those obtained in the mathematical model. Its biggest distribution was observed for *RIC* = 8 μm as in the case of the range of small clearances in the mathematical model. The explanation for the mentioned discrepancy is the increased friction and intermittent contact of rolling-elements corresponding to trapping between two different dynamical states. The level of trapping time both in the mathematical model and for the experiment were similar.

The last considered quantificator, the average length of the diagonal line, had a similar distribution over the radial internal clearance as in the case of the mathematical model. The strongest distribution of the quantificator was observed for *RIC* = 8 μm, and 40 μm similar to the response of the mathematical model. Despite the similar distribution of *L* between the model and experiment, it differed in value, which could be improved by changing the phase-space reconstruction parameters for the mathematical model. 

## 6. Conclusions and Discussion

This paper reported on the impact of the radial internal clearance in rolling-element bearings on its dynamics following the mathematical model and experimental approach. Its impact cannot be studied using standard linear frequency-based methods because the radial internal clearance is strongly connected to the nonlinear contact occurring on the rolling surfaces in the bearing. This fact was the reason why the comparative analysis was conducted between the mathematical model and experiment with the application of recurrence-based methods, with a precise comparison of the results with three different recurrence quantificators. 

The analysis began with the bifurcation analysis of the 2-DOF nonlinear mathematical model of a double-row self-aligning ball bearing and a check of the impact of the radial internal clearance on its dynamics. The analysis showed ranges of optimal clearance based on the minima; however, following the classical approach to the study of the dynamic system, phase-portraits and orbit plots showed a strongly nonlinear character of the bearing. Applying the recurrence analysis to study the mathematical model, three chosen quantificators, i.e., entropy (*ENTR*), trapping time (*TT*), and the average length of the diagonal line (*L*) reflected the qualitative and quantitative changes observed at the bifurcation plots proving its sensitivity to the influence of the radial internal clearance. 

In order to verify the results of the mathematical model, the experimental part was conducted [18] by a few equally distributed values of clearance in the bearing. After the pre-processing of the acceleration time series, the recurrence analysis was conducted for three of the same quantificators as in the case of the mathematical model. The results obtained showed a very similar distribution in the clearance domain and also a similar distribution of the quantificators for specific clearance regions. A wider distribution was observed for the range of small clearances, where fluctuations in the mathematical model were observed. 

Focusing on the outcomes of the conducted research, one of them, without doubt, was finding the range of the optimal clearance during the bifurcation analysis. Given that the radial internal clearance term is part of the equation calculating the deformations occurring on the rolling surfaces related to the contact, it will definitely have nonlinear characteristics. It was expected that low values of deformations and the bearing’s acceleration would be found for low values of clearance and on the other hand, big values of deformations and the bearing’s acceleration were found for higher values of clearance. Nevertheless, it is very rare that a bearing operates by such clearances; sometimes bearings with very small clearances operate at high speeds and under small loads, and sometimes they operate by big values of clearance under heavy loads and low speeds. For most common applications, bearings are produced in the middle range of clearances in the CN and C3 classes, that is why this range of clearances is the biggest area of interest. The results of the bifurcation analysis for the mathematical model showed that the smallest values of acceleration and deformations occurred around the range of clearance of around 30 [μm] What is more, this minima was observed by all studied rotational velocities, so this should be investigated in the experiment as well. The role of the mathematical model in the research was irreplaceable as it showed without any cost the direction in which the experiment should be directed. The experiment was conducted for 10 bearings received from the NTN bearing manufacturer directly from production; the bearings were not prepared on purpose for the reason of the planned experiment. During the experiment, it was important not to have impact from different shape errors, which could affect the response of bearing’s dynamics. That is why for every 10 bearings, it was possible to control the clearance in a wide range with an axial nut and to conduct the experiment by several values of clearance and by different velocities. The experimental results showed a similar distribution of recurrence quantificators as in the mathematical model and the minima of the cubic function was observed in the range of clearance 25–32 μm. The aim of the manuscript was to verify the results of the mathematical model with the experiment. 

To sum up, a comparative analysis of the nonlinear mathematical model of the ball bearing was conducted and experimental data showed the usefulness of the recurrence quantification analysis in the identification of the bearing’s dynamical response by different radial internal clearance. Applied non-linear recurrence indicators were sensitive enough to distinguish between the responses of different systems with respect to the initial condition (defined radial internal clearance) for both the numerical and experimental systems. The next step of the research could be the study of the selection of phase-space reconstruction parameters, which determine the value of quantificators. Nevertheless, the conducted analysis proved the fact that recurrence quantificators can be applied for studying nonlinear time series both for the mathematical model and the experimental data. Another step, which is planned for future research, is the application of additional loads both in the mathematical model and in the experiment to cross-check its impact on the dynamical response of the bearing. Another future direction which should be considered is the implementation of stochastic terms in the mathematical model such as small fluctuations of rotational velocity, shape errors, or clearances in a small range. The results of the research showed the value of an optimal clearance, which was related to small vibrations and a smooth bearing operation. This fact can be taken into account during the durability tests of the bearing, during which the impact of clearance on a bearing’s life has been studied [47]. Nevertheless, this fact is going to be conducted and verified by different loads acting on the bearing, which has the main role during such tests. Additionally, the analysis is going to be conducted by different rotational velocities as in this paper. After obtaining promising results, it is definitely worth continuing the research on the impact of radial internal clearance on bearing dynamics.

## Figures and Tables

**Figure 1 sensors-22-06366-f001:**
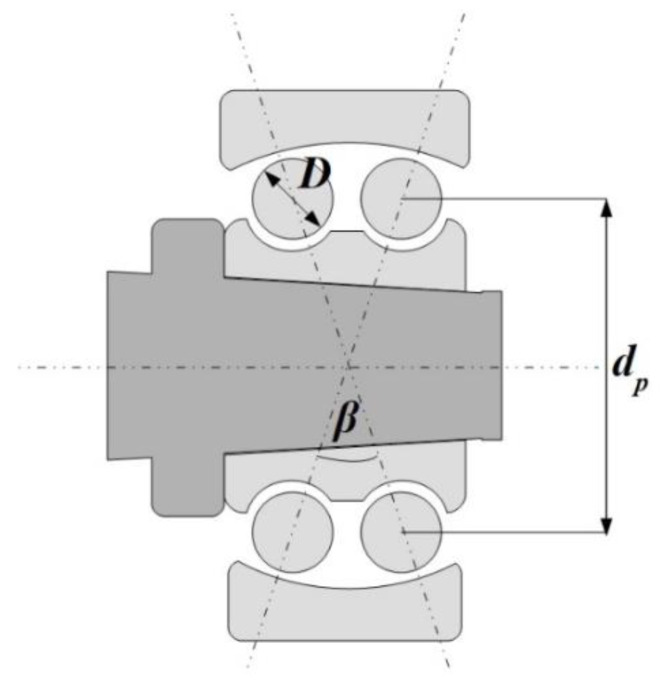
Scheme of the double-row self-aligning ball bearing mounted on the shaft. In the figure, the ball diameter *D*, pitch diameter *d_p_*, and *β* are marked.

**Figure 2 sensors-22-06366-f002:**
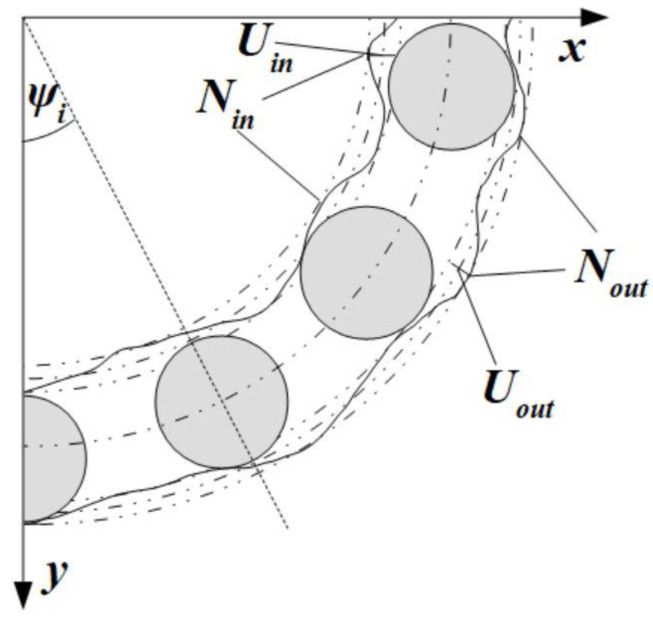
Section of the bearing with marked terms referring to shape errors of the rolling surface *U_in_*, *U_out_*, *N_in_*, *N_out_*.

**Figure 3 sensors-22-06366-f003:**
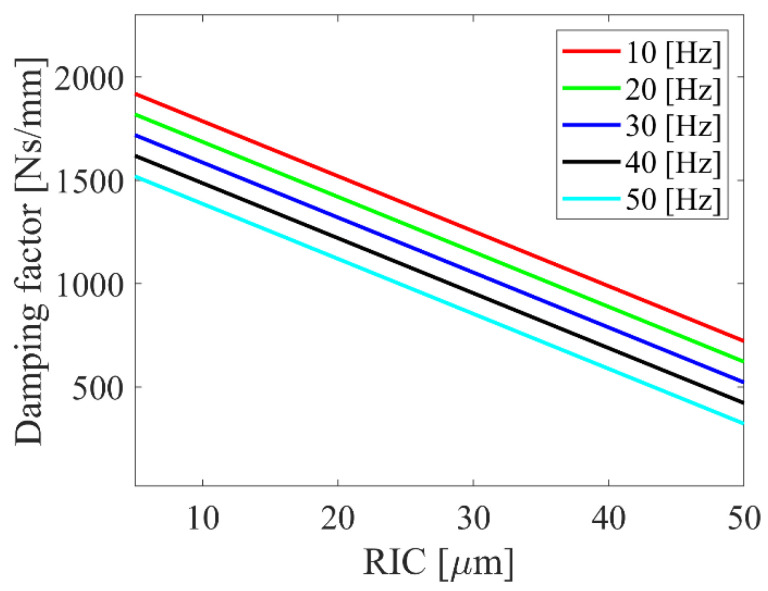
Damping factor functions in the radial internal clearance and rotational velocity domain.

**Figure 4 sensors-22-06366-f004:**
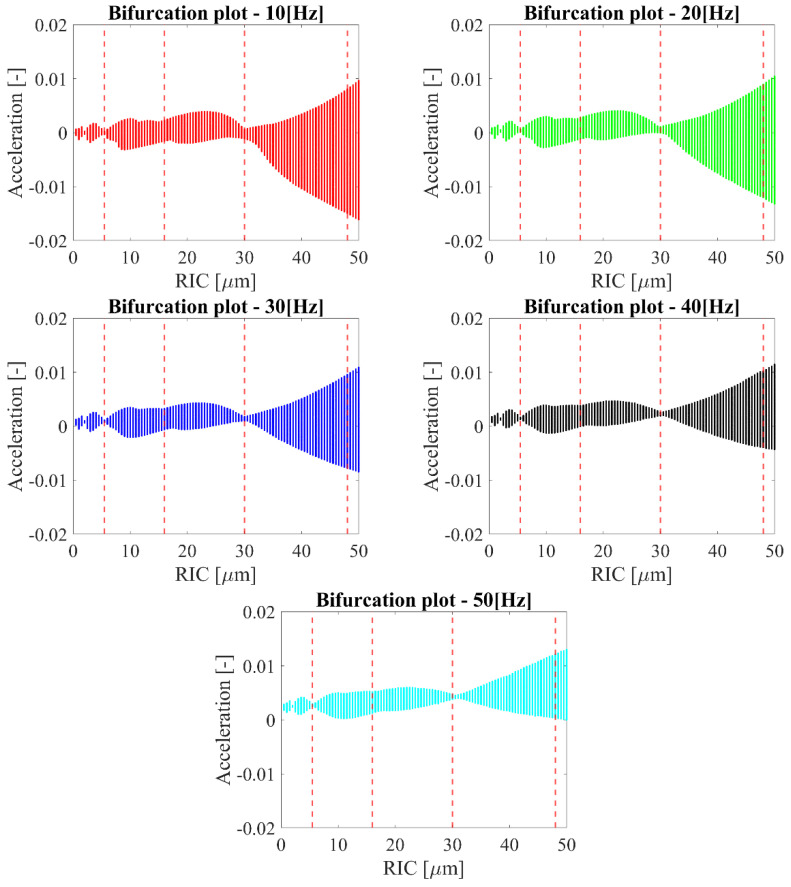
Bifurcation diagrams for different rotational velocities considering during the test.

**Figure 5 sensors-22-06366-f005:**
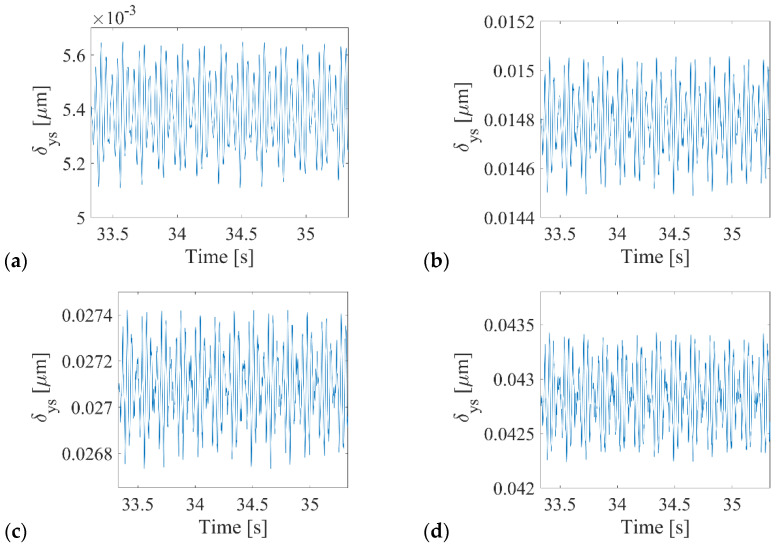
Vertical Deformations *δ_ys_* time series for radial internal clearance (**a**) RIC = 5.5 μm, (**b**) RIC = 16 μm, (**c**) RIC = 30 μm, (**d**) RIC = 48 μm.

**Figure 6 sensors-22-06366-f006:**
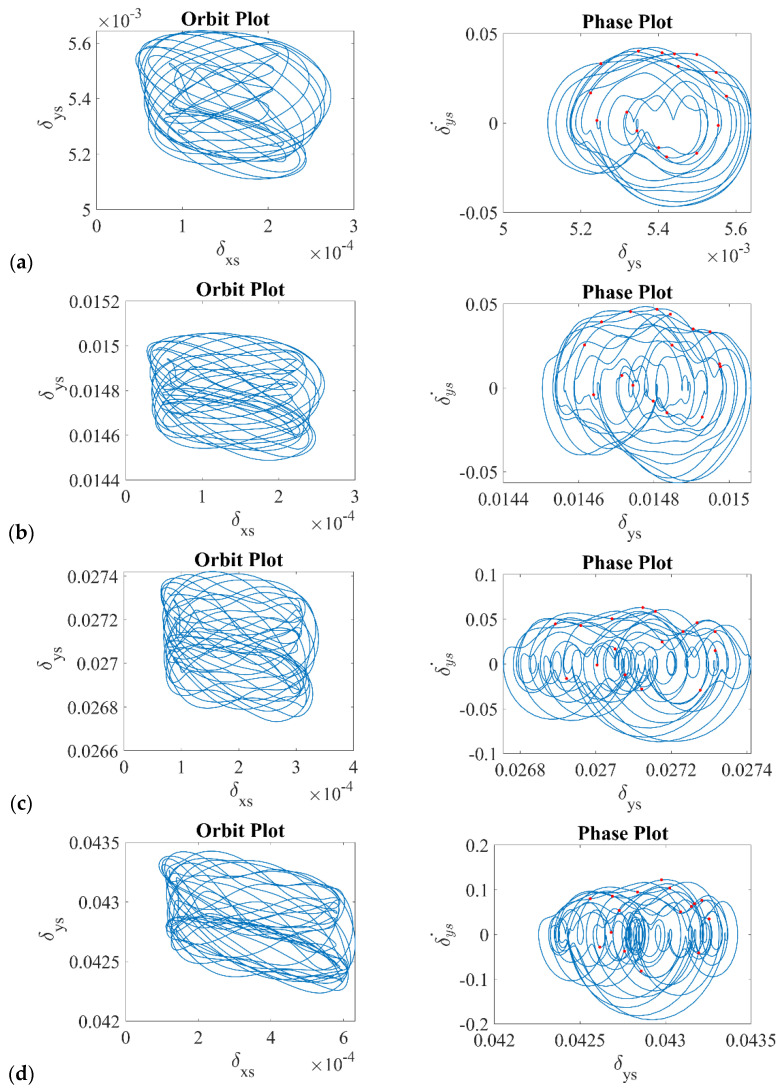
Orbit and phase plots for (**a**) RIC = 5.5 μm, (**b**) RIC = 16 μm, (**c**) RIC = 30 μm, (**d**) RIC = 48 μm.

**Figure 7 sensors-22-06366-f007:**
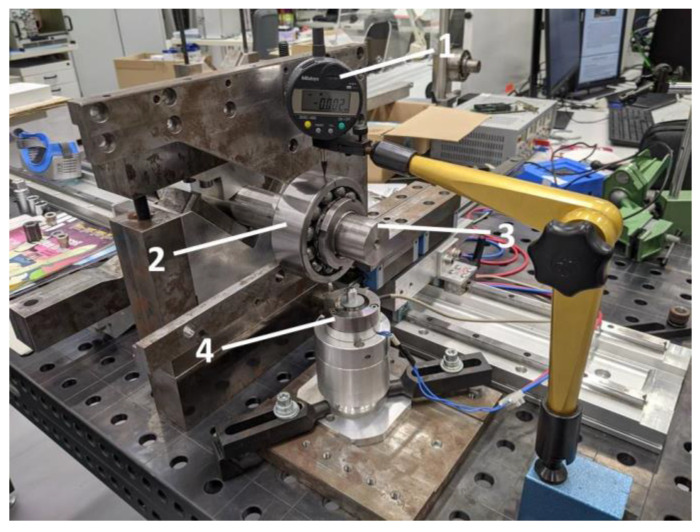
Automated system with the automated radial internal clearance measurement after mounting with the studied bearing NTN 2309SK. 1—Dial gauge, 2—Seated bearing, 3—Shaft, 4—Electromagnetic actuator and holder.

**Figure 8 sensors-22-06366-f008:**
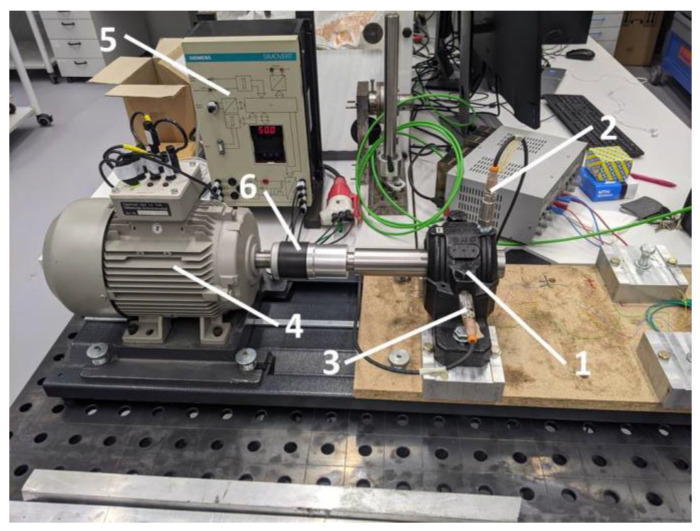
Experimental setup: 1—Plummer block with mounted ball bearing, 2—Vertical accelerometer, 3—Horizontal accelerometer, 4—Electric motor, 5—Inverter, 6—Coupling.

**Figure 9 sensors-22-06366-f009:**
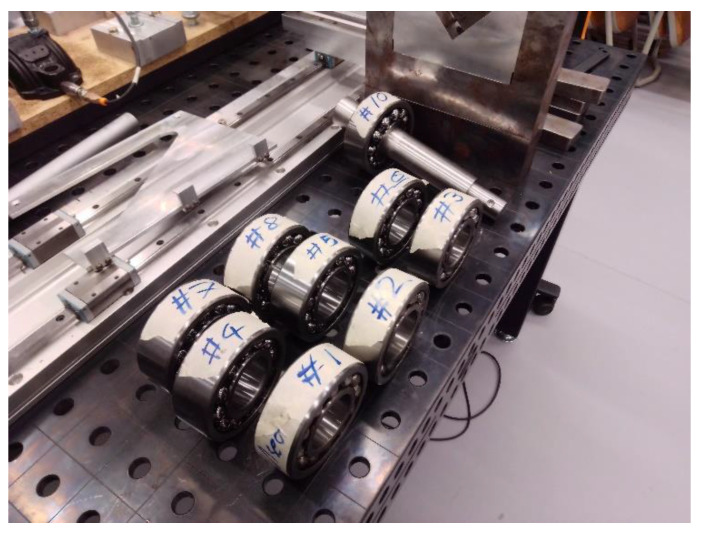
Bearings studied in the experiment.

**Figure 10 sensors-22-06366-f010:**
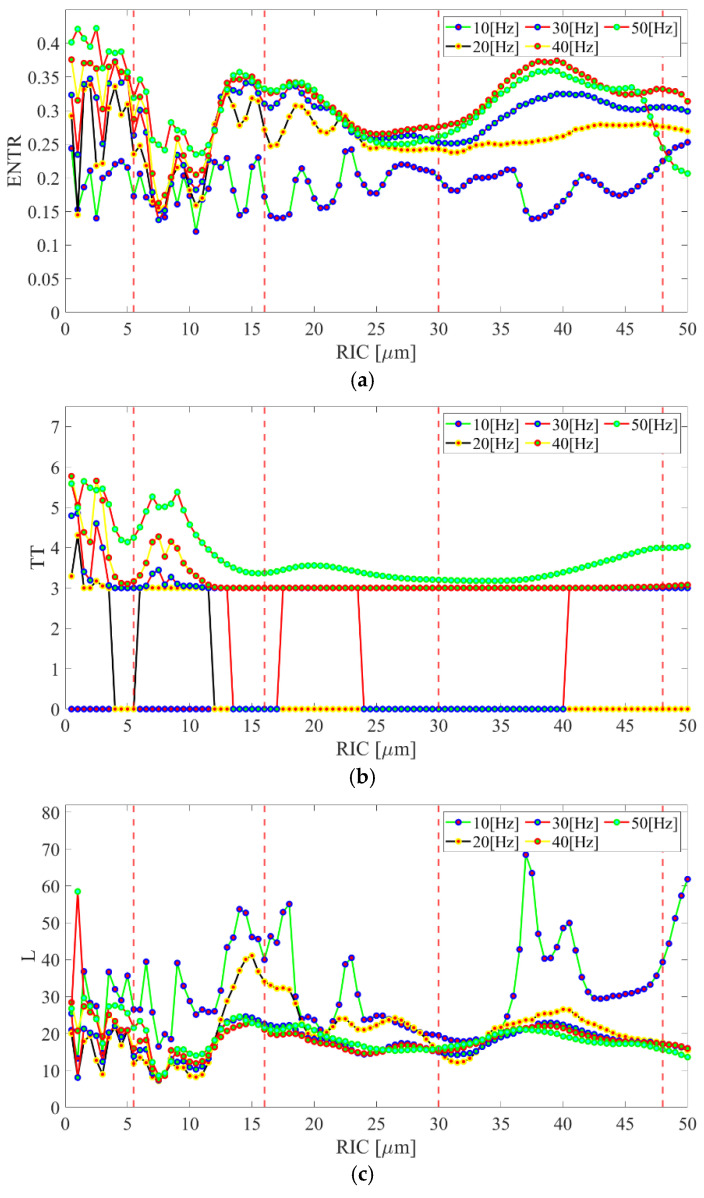
Results of the recurrence quantificators for the results obtained from mathematical model: (**a**) Entropy*—ENTR*, (**b**) Trapping Time—*TT*, (**c**) Average length of the diagonal line—*L*.

**Figure 11 sensors-22-06366-f011:**
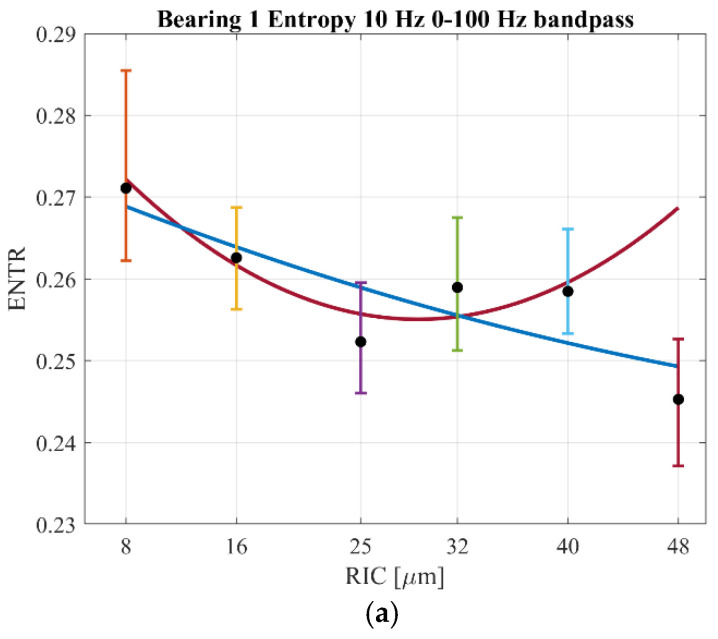
Results of the recurrence quantificators for the results obtained from the experiment for Bearing 1: (**a**) Entropy*—ENTR*, (**b**) Trapping Time—*TT*, (**c**) Average length of the diagonal line—*L* [20]. The blue line refers to the entire range of considered clearance values, while the red cubic function refers to the fitting to four middle clearance values.

**Table 1 sensors-22-06366-t001:** Terms applied for the mathematical model of the ball bearing.

Term	Symbol	Unit	Value
Rotational velocity	*ω_s_*	Hz	10; 20; 30; 40; 50
Ball diameter	*D*	mm	15.875
Pitch diameter	*d_p_*	mm	71.810
Pressure angle	*β*	°	15.520
Radial Internal Clearance	*r_c_*	μm	0.5:0.5:50
Load deflection factor	*K*	N/mm	1,874,580
Damping factor in the vertical/horizontal direction	*c_x_*, *c_y_*	Ns/mm	725.2–2029.3 (10 Hz) 625.2–1929.3 (20 Hz) 525.2–1829.3 (30 Hz) 425.2–1729.3 (40 Hz) 325.2–1629.3 (50 Hz)
Mass of the inner ring and shaft	*m*	kg	1
Amplitude of the biggest undulation on the inner ring	*U_in_*	mm	0.0004
Amplitude of the biggest undulation on the outer ring	*U_out_*	mm	0.00028
Number of undulations on the perimeter of the inner ring	*N_in_*	-	28
Number of undulations on the perimeter of the outer ring	*N_out_*	-	19
Number of rolling elements	*n*	-	26
Eccentricity	*e*	μm	0.35–(10 Hz) 0.80–(20 Hz) 0.71–(30 Hz) 0.59–(40 Hz) 0.65–(50 Hz)
External vertical force	*F_y_*	N	50
External horizontal force	*F_x_*	N	0

**Table 2 sensors-22-06366-t002:** Phase-space reconstruction parameters selected for the mathematical model and experimental data.

Term	Time Delay (*τ*)	Embedding Dimension (*m*)	Threshold (*ε*)
Mathematical model	5	5	Based on the constant *RR*
Experimental data	7–10	3	

## Data Availability

Data available on request due to privacy restrictions.

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
