# Peer review of "Experimental Verification of the Impact of Radial Internal Clearance on a Bearing’s Dynamics"

_sensors, 2022, doi:10.3390/s22176366_

Round 1

Reviewer 1 Report

The paper contribution is the comparison between mathematical model of the ball bearing and experimental data. The results showed the identification of bearing’s dynamical response by different radial internal clearance.

The manuscript is well prepared and it contains the expected structure of scientific paper. The manuscript has a systematic presentation of theoretical and experimental part of research. However, the reviewer is missing a critical assessment of presented results and some guidelines for practical application of the results (there is no discussion section). Can you specify some generalized guidelines how to select radial clearance for different applications? What is relation between radial clearance and bearings performance (expected lifespan, noise level, etc.).  

Please, find below some recommendations for the paper improvement:

Add explanation of  2-DOF model. Check the other abbreviations too.

There are some typo (repeated text). Check page 2

The model is based is based on Hertzian Contact Theory 

Figure 11 – add a legend; what is difference between blue and red line? Figure a), b), and c).

How the samples (bearings with different clearance) has been prepared. Have they been selected from the serial production or they have been manufactured for tests only with the target clearance? 

Author Response

Lublin, 18.08.2022

Response to Reviewer no. 1

Dear Reviewer,

We appreciate the constructive suggestions enclosed in your review. All the authors of this manuscript are grateful for Reviewer’s encouraging approach, as well as constructive criticism that is intended to improve our work. We carefully considered the comments
and herein we explain how we revised the manuscript based on your recommendations.

It is our belief that the manuscript is substantially improved after making the suggested edits. We want to express our appreciation for taking the time and effort necessary to provide guidance.

Yours sincerely

Bartłomiej Ambrożkiewicz

Lublin University of Technology

Poland

Reviewer 2 Report

I recommend publishing the paper in the Sensors journal after minor corrections. On page 2, line 88, the phrase "is based" is repeated twice. Decipher the abbreviation RIC at the first mention of it. On page 6, line 220, in table 1, it may be clearer to indicate the rotational velocities in Hz near the eccentricity value in brackets. On page 10, line 277, it might be better to remove the notation 1). On page 10, line 297, eddyNCDT must be separated by a space.

Author Response

Lublin, 18.08.2022

Response to Reviewer no. 2

Dear Reviewer,

We appreciate the constructive suggestions enclosed in your review. All the authors of this manuscript are grateful for Reviewer’s encouraging approach, as well as constructive criticism that is intended to improve our work. We carefully considered the comments
and herein we explain how we revised the manuscript based on your recommendations.

It is our belief that the manuscript is substantially improved after making the suggested edits. We want to express our appreciation for taking the time and effort necessary to provide guidance.

Yours sincerely

Bartłomiej Ambrożkiewicz

Lublin University of Technology

Poland

Reviewer 3 Report

Report on the Manuscript ID: Sensors-1871797

Title: Experimental verification of radial internal clearance impact on bearing's dynamics

This work investigates the influence of radial internal clearance on the dynamics of rolling-element bearing. The analysis showed characteristic dynamical behavior by specific clearance range, what reflects among others in low or high amplitude, that can refer to the optimal clearance.

The paper is well written and it is suitable to the submitted Journal. However, I have the following suggestions:

1.    The paper should be carefully double-checked from grammatical point of view.

2.    Nomenclature of the used symbols must be added to the paper.

3.    The introduction of the paper (Section 1) should be expanded and improved. In my opinion, Section 1 (Introduction) should be subdivided into five brief Subsections: 1.1) Background, 1.2) Formulation of the Problem of Interest for this Investigation, 1.3) Literature Survey, 1.4) Scope and Contribution of this Study, 1.5) Organization of the Paper as stated in the end part of introduction section.

4.    Authors should improve the introduction by including the recent development within the frame of the corresponding analytical solutions with their illustrative applications by considering the help of recently published papers.

·         https://doi.org/10.1007/s11071-022-07722-x

·        https://doi.org/10.1016/j.ymssp.2020.107138

5.    Resolution of most of figures is very poor.

6.    Have the authors employed any assumptions to deal with the investigated model? Please explain briefly.

7.    Concluding Remarks section must be extended so that it provides and covers all the finding of the paper and future direction. Moreover it should mention some important meaning of simulations as conclusion.

Author Response

Lublin, 18.08.2022

Response to Reviewer no. 3

Dear Reviewer,

We appreciate the constructive suggestions enclosed in your review. All the authors of this manuscript are grateful for Reviewer’s encouraging approach, as well as constructive criticism that is intended to improve our work. We carefully considered the comments
and herein we explain how we revised the manuscript based on your recommendations.

It is our belief that the manuscript is substantially improved after making the suggested edits. We want to express our appreciation for taking the time and effort necessary to provide guidance.

Yours sincerely

Bartłomiej Ambrożkiewicz

Lublin University of Technology

Poland

Round 2

Reviewer 3 Report

Resolution of most of figures is still very poor.

Author Response

Dear Reviewer,

We appreciate the constructive suggestions enclosed in your review. All the authors of this manuscript are grateful for Reviewer’s encouraging approach, as well as constructive criticism that is intended to improve our work. We carefully considered the comments
and herein we explain how we revised the manuscript based on your recommendations.

It is our belief that the manuscript is substantially improved after making the suggested edits. We want to express our appreciation for taking the time and effort necessary to provide guidance.

Yours sincerely

Bartłomiej Ambrożkiewicz

Lublin University of Technology

Poland
